# Circum-Antarctic bottom water formation mediated by tides and topographic waves

Xianxian Han [1,2], Andrew L. Stewart [3], Dake Chen[1,2,4] ✉, Markus Janout [5], Xiaohui Liu[4], Zhaomin Wang[1] & Arnold L. Gordon [6]

The downslope plumes of dense shelf water (DSW) are critical for the formation of Antarctic Bottom Water (AABW), and thus to the exchange of heat and carbon between surface and abyssal ocean. Previous studies have shown that tides and overflow-forced topographic Rossby waves (TRWs) may have strong impact on the downslope transport of DSW, but it remains unclear how the combined action of these two processes influence the descent processes of DSW, and of the resulting AABW properties. Here, with a synthesis of historical in situ observations and a set of numerical model experiments, we show that tides and TRWs play comparable roles in AABW formation: they both act to accelerate DSW descent to the abyss, leading to the formation of colder and denser AABW. Yet, tides have little impact on AABW formation unless the continental slope is steep enough to suppress TRW generation. We further characterize the dynamical regimes of dense overflows around the entire Antarctic continent based on the relative importance of TRWs versus tides. These findings highlight the pervasive role of high-frequency processes, which are not well represented in the present climate models, in the formation of AABW, and thus in the global overturning circulation.

Antarctic Bottom Water (AABW) originates from the descent of dense shelf waters (DSW), which are formed via a combination of brine rejection during sea ice growth and ocean/ice shelf interactions around Antarctic coast[1,2]. Due to its relatively high density, DSW overflows across the shelf break, descends over the continental slope, entraining ambient water, to produce AABW[3,4]. This process ventilates abyssal ocean and impacts the heat and carbon inventory, and thus plays a key role in global ocean overturning circulation and climate[5–7].

It has long been established that the Weddell and Ross Seas are the primary source regions of AABW[8–10]. In recent years, the Adélie coast region and Prydz Bay have also been identified as areas of AABW formation[11–13]. A review of nearly 50 years of direct hydrographic observations show that dense overflows are widely spread around Antarctic continental margin[14] (Fig. 1a), all potentially contributing to the formation of AABW. Motivated by the increasing number of different situations in which AABW is formed, in this study we seek a general understanding of the dynamical mechanisms that govern the descent of DSW, and their influences on the entrainment that transforms DSW into AABW.

Theoretically, under the strong influence of the Earth's rotation in polar regions, a steady downslope flow should turn westward to spread approximately along isobaths, while descending slowly via benthic layer Ekman transport[15,16]. Variable bathymetry may facilitate this downslope flow via topographic steering[2,13,17,18]. However, in situ observations show that DSW can also reach the

[1]Southern Marine Science and Engineering Guangdong Laboratory (Zhuhai), Zhuhai, China. [2]School of Atmospheric Sciences, Sun Yat-sen University, Zhuhai, China. [3]Department of Atmospheric and Oceanic Sciences, University of California, Los Angeles, CA, USA. [4]State Key Laboratory of Satellite Ocean Environment Dynamics, Second Institute of Oceanography, Ministry of Natural Resources, Hangzhou, China. [5]Alfred Wegener Institute, Helmholtz Centre for Polar and Marine Research, Bremerhaven, Germany. [6]Lamont Doherty Earth Observatory, Columbia University, Palisades, NY, USA. ✉e-mail: dchen@sio.org.cn

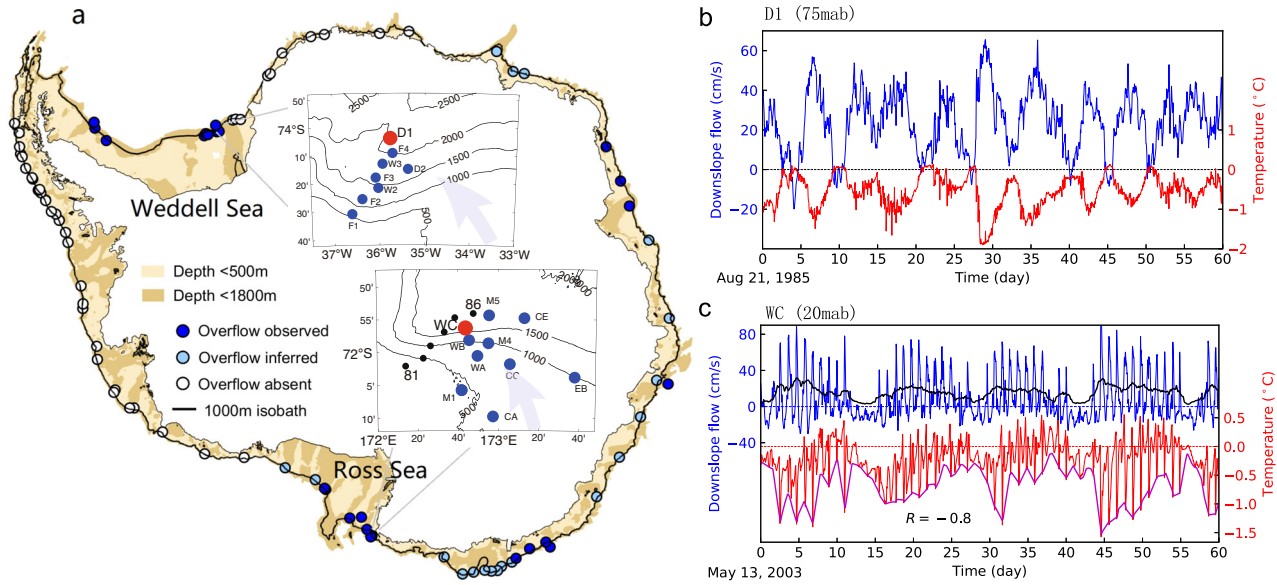

**Fig. 1 | Observations of dense overflows around Antarctica. a** Observed overflow sites around the Antarctic coast[14]. The insets indicate the Weddell Sea and Ross Sea overflow sites, respectively. Black dots in the Ross Sea inset correspond to Conductivity–Temperature–Depth profiler stations 81–86, used in Fig. 2. The red dots indicate the moorings shown in (**b**, **c**), and blue dots in the insets denote other available moorings. **b**, **c** Time series of temperature and velocity for moorings D1 (75 m above bottom (mab), Weddell Sea) and WC (20 mab, Ross sea), respectively. The thick black curve in (**c**) indicates the fitted daily tidal flow strength (see "Methods"), while the thick magenta curve indicates the daily minimum temperature. The correlation between the two curves is −0.8.

deep ocean over a relatively short along-slope distance, even in the absence of topographic steering[12,19–21]. This implies that other mechanisms must serve to accelerate the descent of DSW. Two such mechanisms have been identified in previous studies: tidal effects[22–25] and topographic Rossby waves (TRWs) associated with baroclinic instability of dense overflows[16,26–29]. However, it remains unclear to what extent tides versus TRWs influence the DSW downslope transport and entrainment, and thus the formation rate and properties of AABW, across the range of overflow regimes found around Antarctic margins.

Here, we synthesize historical in situ data in the Weddell Sea and Ross Sea, where observations are relatively abundant, and formulate an idealized high-resolution process-oriented numerical model to investigate the roles of tides and TRWs on the formation of AABW. We explore their effects across a parameter range wide enough to span the behaviors around the entire Antarctic continental margin, and generalize our findings to give a pan-Antarctic description of the dynamical regimes in different overflow sites.

## Results

We first identify the modes of variability observed in Antarctic DSW overflows, using moored measurements from the two most heavily-instrumented overflow sites (see Methods). Both the Weddell Sea (Fig. 1b) and the Ross Sea (Fig. 1c) exhibit significant downslope flows with large periodic oscillations over the lower continental slope. However, there is a fundamental difference between the Ross and Weddell Seas: the oscillations in the Ross Sea are mainly diurnal, consistent with tidal fluctuations[23,30], while those in the Weddell Sea have a primary periodicity of several days, consistent with TRWs[29,31]. However, despite the different periods and processes, both sites exhibit a close correspondence between the oscillations of the downslope flow and the water temperature. This suggests that the downslope transport of DSW is mediated by tides in the Ross Sea, and by TRWs in the Weddell Sea.

In addition to diurnal tidal oscillations, the Ross Sea also shows a spring-neap fortnightly tidal cycles with a period of approximately 14 days. The coldest waters emerge during the spring tide rather than

the neap tide; the correlation between the daily tidal flow strength (black line in Fig. 1c, see "Methods") and minimum temperature (magenta line) is −0.8. The fact that the coldest waters emerge during the spring tide therefore suggests that temperature variability is governed by the tendency of the tides to advect DSW across the continental slope[22,23], rather than by the mixing induced by the tidal flow[32,33]. This phenomenon is robust across all the moorings on the continental slope and is further supported by other moorings farther seaward of the continental slope (Supplementary Fig. 1). In contrast, stronger tidal flow dilutes DSW on the continental shelf break (moorings M1 and CA, Supplementary Fig. 1), consistent with the influence of stronger tidal mixing[32,33].

### Tidal influence on Antarctic overflows

To interpret how tides and TRWs control the rate of DSW descent and entrainment, we use an eddy-resolving process-oriented numerical model with 500 m horizontal resolution in the vicinity of the DSW overflow[34–36], and track the overflow with passive tracers (see "Methods" and Supplementary Fig. 2 for detailed model setup). The results show that in a Ross Sea parameter regime, tracers are advected to greater depths and retain higher concentration in the presence of tides (Fig. 2a, b), though the tracer concentrations are more diluted in the regions shallower than 500 m depth (Fig. 2c, d). Hence, tides actually reduce the rate at which DSW is mixed with overlying waters on the continental slope: zonally integrated tracer diapycnal mass flux is temporally steady and substantially stronger in the experiment without tides (Fig. 2c), but becomes weaker and exhibits periodic variations when tides are included (Fig. 2d). Figure 2e shows a large variation of the pycnocline depth in the tidal experiment—deepening of the pycnocline coincides to the downslope transport of DSW—while the pycnocline depth remains nearly steady in the absence of tides. The pycnocline deepening is induced by the flow convergence during the ebb tide (Supplementary Fig. 3), which produces a V-shaped frontal structure that drives the cold, fresher surface waters to descend to over 1000 m. In situ observations display a similar structure (Fig. 2f, g). Therefore, the tidal flow has two key effects on the DSW overflow: one is to pump the DSW quickly to the deep ocean and reduce

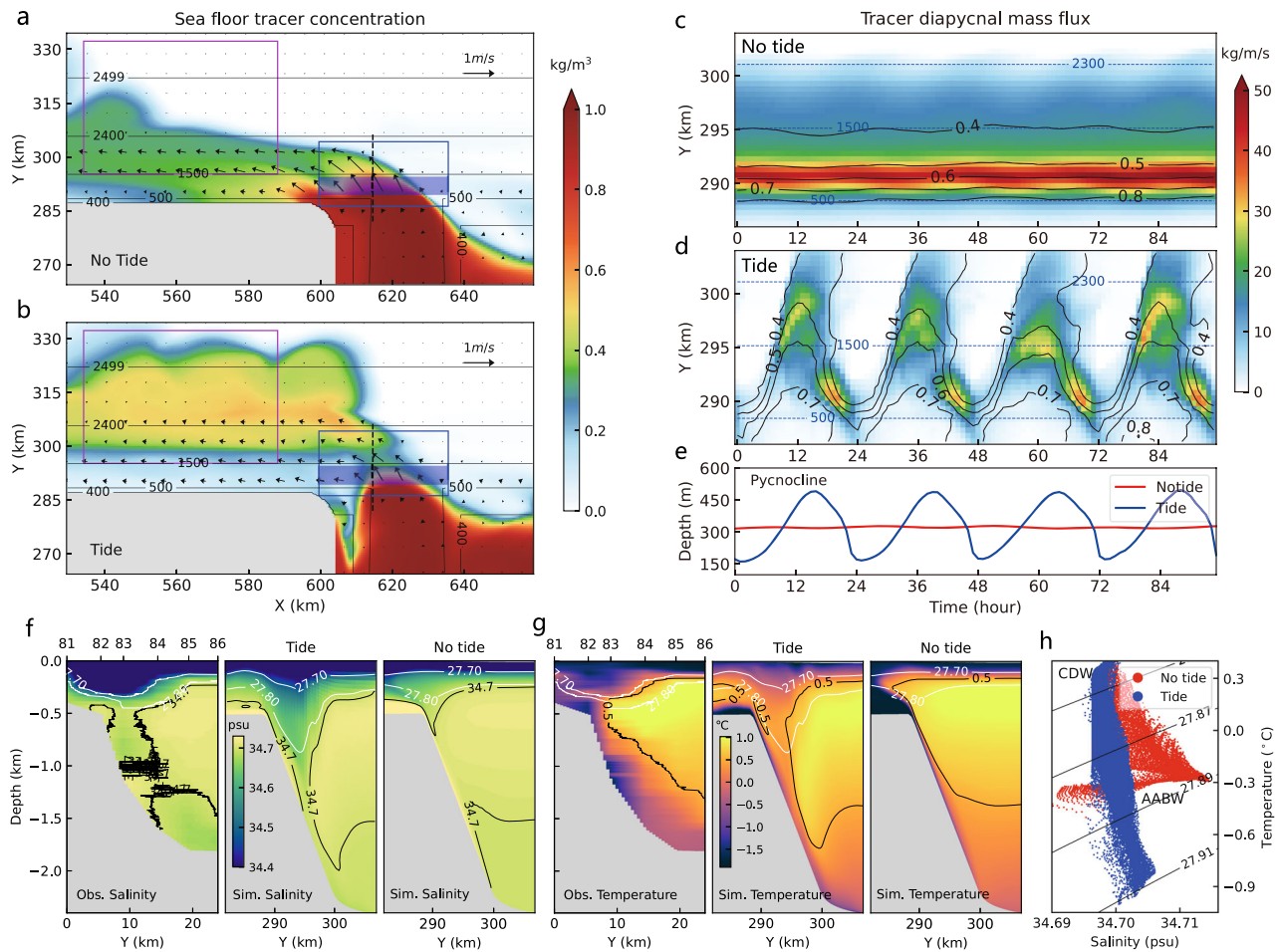

**Fig. 2 | Observations and idealized numerical experiments of the Ross Sea overflow.** 10 day-averaged sea floor tracer concentration and flow speed/direction (arrows) for experiments **a** without and **b** with K1 tidal forcing. The magenta boxes indicate the regions used for (**h**), and the blue boxes indicate the regions for (**c**, **d**). **c**, **d** Hovmüller diagrams of zonally-integrated diapycnal (27.88 kg/m³, with 1000 kg/m³ subtracted, referenced to surface) tracer mass flux. The black contours indicate the mean tracer concentration in the dense overflow layer, and the blue contours correspond to isobaths. **e** Time series of pycnocline depths in experiments with and without tides. The pycnocline depth is defined as the depth of density interface of 27.80 kg/m³, averaged over the light blue-shaded region in (**a**, **b**). **f** Observed (Obs.) salinity section that cross the continental slope indicated in Fig. 1a, and the snapshots of simulations (Sim.) with and without tidal forcing along the sections shown by the black dashed line in (**a**, **b**). The white and black contours indicate potential density (kg/m³) and salinity (psu) respectively. **g** Similar to (**f**), but for potential temperature. **h** 10 day-averaged T-S diagram below 1500 m depth, drawing data from the regions of magenta boxes shown in (**a**, **b**).

entrainment; the other is to induce the V-shaped front that permits the DSW to entrain relatively cold, low salinity water[37], instead of warm and salty circumpolar deep water (CDW) (Fig. 2f, g). This could explain the colder but fresher AABW in both observations and tidal experiment (Fig. 2f–h and Supplementary Fig. 1b).

## Overflow-forced TRWs

In contrast to the Ross Sea overflow, observations in the Weddell Sea show that overflow-forced TRWs are dominant and exhibit two clusters of frequencies, with higher and lower frequencies concentrated on the upper and lower continental slope respectively, albeit with some overlap[31,38] (Fig. 3a, b). Tidal flow in the Weddell Sea is strong near the shelf break region, but is relatively weak compared to the Ross Sea, and becomes much weaker on the continental slope (Fig. 3b). Taking 20 cm/s as a representative maximum cross-slope tidal flow speed at 1000 m isobath[39], the estimated cross-slope excursion (see "Methods") of the diurnal tide is about 5 km, and the corresponding vertical displacement is less than 250 m, suggesting that tides cannot produce efficient downslope transport in this region. In contrast, the stronger and lower-frequency oscillations associated with TRWs can be expected to produce larger excursions that facilitate the descent of DSW.

Based on these considerations, we first neglect tides in our Weddell Sea-like experiment to investigate the role of TRWs exclusively. The model setup is similar to the Ross Sea experiment (see "Methods"), except for a spatially-varying topography to approximate the Weddell Sea continental slope. Although the setup is rather simplified and excludes tides and the inflow of the Antarctic Slope Current (ASC) from the east, the model is able to simulate TRWs that qualitatively resemble the observations (Fig. 3a–d). The downslope transport of DSW coincides with the offshore velocity associated with the TRWs (Fig. 3c), while isolated eddies form in the regions shallower than 1500 m, and confine boluses of DSW to the upper continental slope[16]. We explain the downslope transport from the perspective of energy conservation: the only energy source in our model is the potential energy input associated with DSW formation on the continental shelf, and this potential energy is released as the DSW slides into the deep ocean, energizing TRWs[16,40]. Similar to tides, TRWs can also induce a V-shaped front near the shelf break (Fig. 3e). Note that this frontogenesis occurs in the absence of winds[35,41], tidal forcing[42,43], and coastal freshwater inputs[44]. Thus TRWs may have similar effects on DSW overflows as tides, reducing entrainment and produce colder and fresher AABW. However, this cannot be demonstrated directly in our model because TRWs are internally generated and it is impossible to

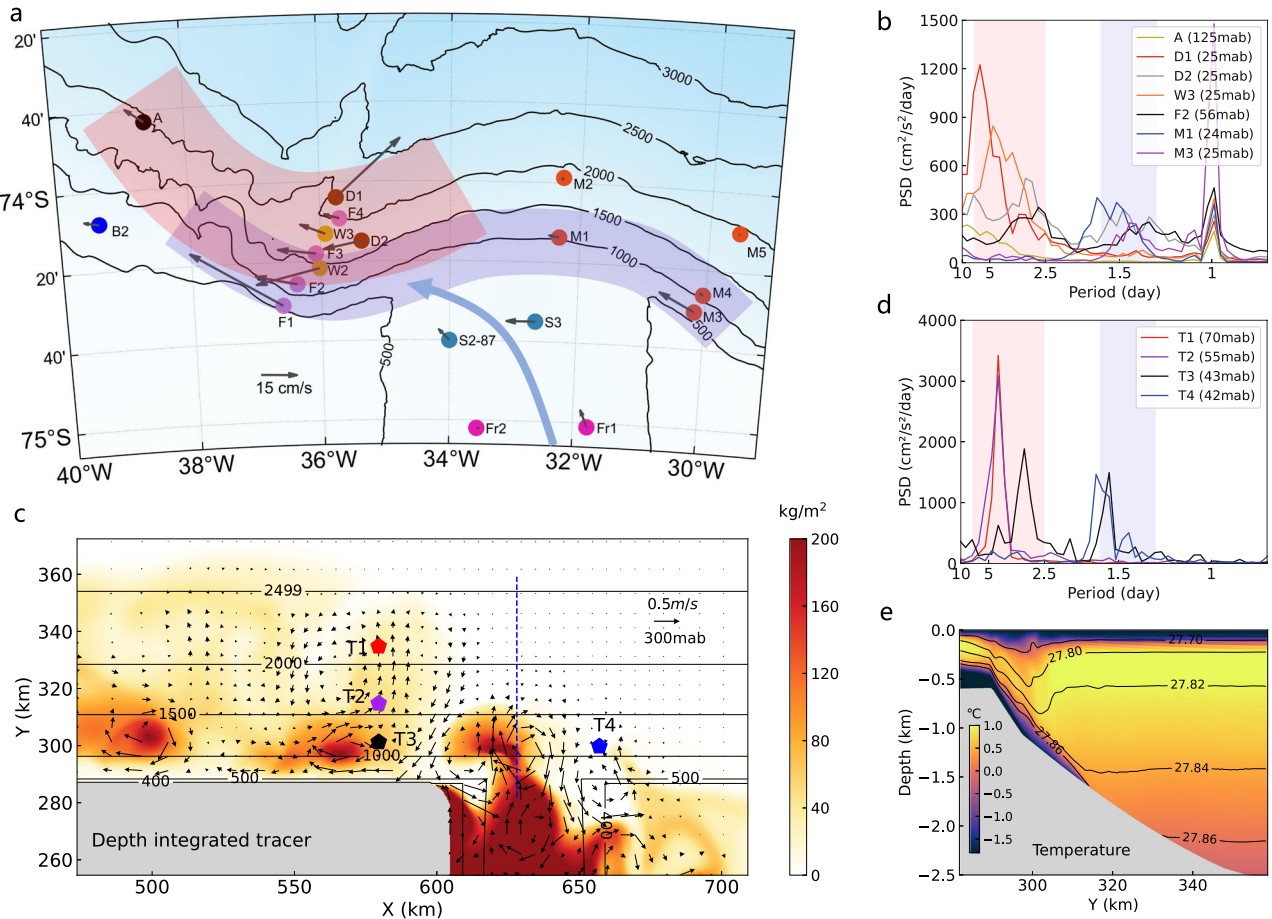

**Fig. 3 | Observations and simulations of the Weddell Sea overflow. a** Historical mooring locations, with colored dots representing different sets of moorings. Arrows indicate time-averaged flows close to the sea floor. The light purple and red shadings on the continental slope indicate the approximate spatial ranges of topographic Rossby waves (TRWs) with different frequencies (see (**b**, **d**)). **b** Power spectral density (PSD) of velocity for several representative moorings shown in (**a**). **c** A snapshot of vertically integrated tracer concentration in the Weddell-like model simulation, with the arrows at 300 mab (meters above bottom) denoting the circulations of TRWs. The colored pentagons indicate the sites used for spectral analysis in (**d**). **d** PSD of velocity for the selected sites shown in (**c**). Note the different *y*-axis range from (**b**). **e** Cross-slope/vertical distribution of potential temperature along the section indicated by the blue dashed line in (**c**).

create a comparative experiment without TRWs for the Weddell-like model configuration.

## Combined effects of tides and TRWs

According to previous studies, dense overflows are intrinsically unstable and can energize coupled TRWs that oscillate throughout the water column, but such an instability is suppressed by steep bathymetric slopes[26,29]. This is consistent with above analysis (Figs. 1–3) showing that a Ross Sea-like overflow (characterized by a large slope steepness, $s = \frac{\Delta H}{\Delta L} = 0.15$) exhibits no energy at TRW frequencies, while TRWs are prevalent in a Weddell Sea-like overflow where the slope is relatively gentle ($0.02 \lesssim s \lesssim 0.07$). These findings indicate that the bathymetric slope plays a key role in delineating different dynamical regimes of overflow[29]. However, it remains unclear to what extent tides and TRWs interact with each other and jointly influence the downslope transport of DSW.

To investigate the combined effects of tides and TRWs in different parameter regimes, we conduct a series of sensitivity experiments with varied slope inclines to simulate a range of overflow conditions around Antarctica (see Methods and Supplementary Fig. 4 and Table 1). We track the depth to which DSW has descended by finding the isobath corresponding to the DSW tracer's center of mass, $H_{DSW}$, as a function of along-slope distance (see Methods). This

diagnostic allows us to visualize the mean descent pathway of DSW (Fig. 4a). In experiments without tides, the rate at which DSW descends to the deep ocean varies non-monotonically with slope steepness. This is due to the transitions among different dynamical regimes as the slope steepness increases, with a steady, quasi-along-slope overflow when the slope is relatively steep, formation of TRWs at intermediate slope steepness, and genesis of nonlinear eddies for gentle slopes[16]. In Fig. 4b we highlight this non-monotonic dependence on slope steepness by averaging $H_{DSW}$ between $x = 585$ km and $x = 595$ km. When tides are added, DSW is advected to the deep ocean more rapidly in experiments with relatively steep bathymetric slopes ($s \geq 0.125$), whereas for gentler slopes ($s \leq 0.1$) the inclusion of tides has little impact on the descent pathway (Fig. 4a, b). This is because TRWs are suppressed over steep slopes and thus tides are needed to push DSW to the deep ocean, whereas gentle continental slopes have an offshore width far exceeding the tidal excursion distance, and the development of TRWs serves as the primary driver to accelerate DSW downslope.

As has been discussed above, our simulations indicate that the downslope advection of DSW by tides reduces the cumulative entrainment of overlying waters into DSW as it descends (Fig. 2), and that TRWs might have a similar effect. To quantify the impact of tides on entrainment, we now diagnose the probability density function for

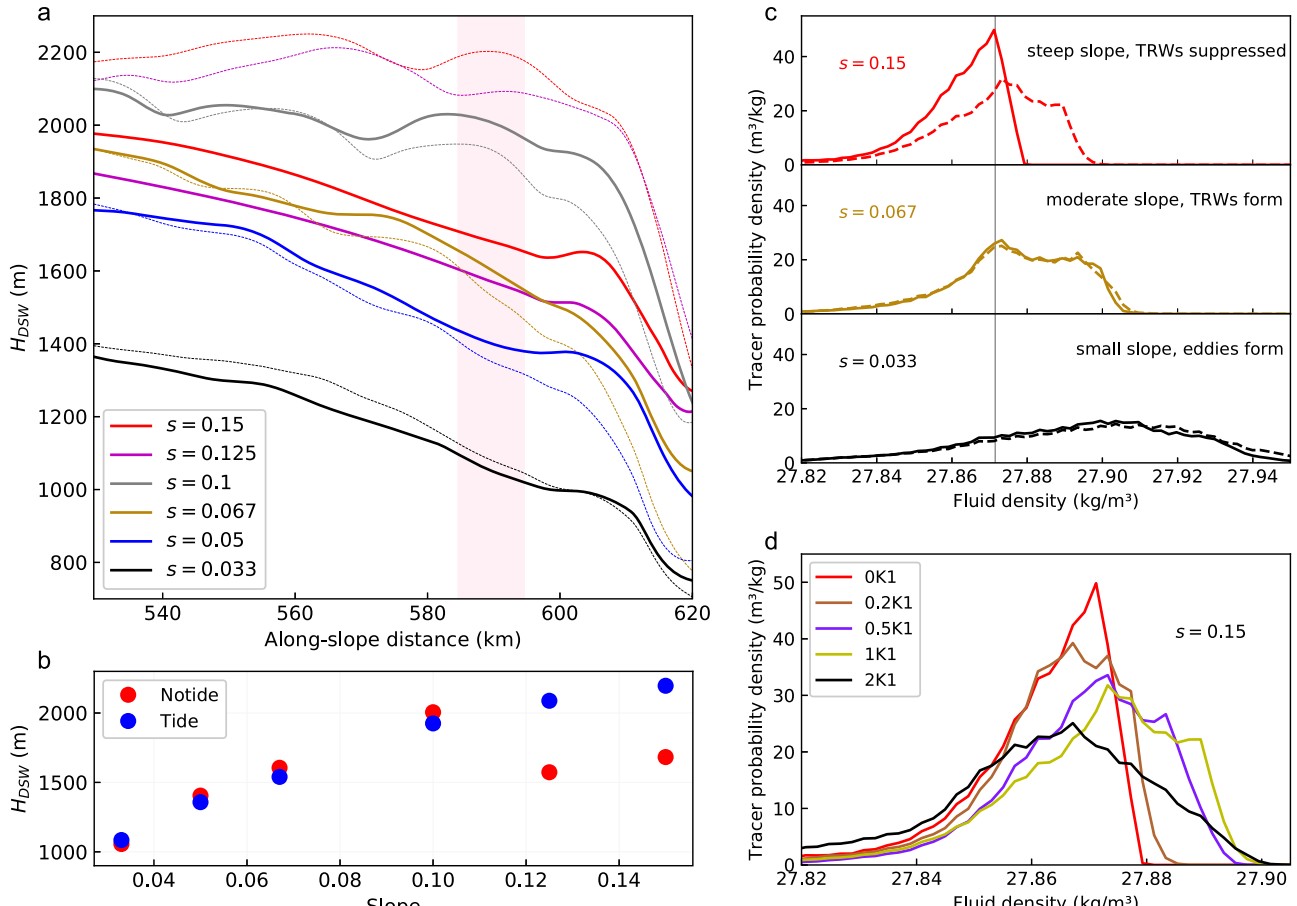

**Fig. 4 | Simulations investigating the influences of tidal flows and continental slope steepness on dense overflows. a** Isobath corresponding to the dense shelf water (DSW) tracer center of mass, as a function of along-slope distance downstream of the trough. The solid thick curves indicate the simulations without tidal forcing, while the dashed curves indicate the corresponding simulations that include tidal forcing. The semi-transparent red shading denotes the zonal range that we calculate the averaged DSW tracer-weighed isobath shown in (**b**). **b** Zonally averaged DSW tracer-weighted isobath for different experiments, with and without tidal forcing. **c** Probability density function of tracer as a function of potential density for experiments with three different slope inclines, with and without tidal forcing, computed over the area downstream of the semi-transparent red shading shown in (**a**). The gray vertical line indicates the maximum potential density in the simulation prior to DSW production. **d** Similar to (**c**), but for varying tidal forcing strengths with constant slope steepness ($s = 0.15$).

the tracer as a function of the potential density:

$$p_i = \frac{1}{\Delta\rho} \frac{\overline{\iiint_{B_i} \tau dv}}{\overline{\iiint \tau dv}}. \tag{1}$$

Here $p_i$ is the probability that tracer lies in the $i^{th}$ density bin, which we denote as $B_i \equiv \rho_i - \frac{\Delta\rho}{2} < \rho_i < \rho_i + \frac{\Delta\rho}{2}$. The overbar indicates 10 days mean, and the density bin size is $\Delta\rho = 0.002$ kg/m³. Figure 4c presents three experiments with bathymetric slopes that are steep (suppressing TRW formation), moderate (permitting TRWs) and gentle (forming nonlinear eddies), respectively (see Supplementary Fig. 5 for all experiments). Among these three experiments, DSW is lightest, and thus most strongly diluted relative to its properties on the shelf, when the slope is steep ($s = 0.15$). Adding tides shifts the tracer distribution to greater densities, indicating that less entrainment is occurring as DSW descends, resulting in denser AABW being formed at the bottom of the slope. In contrast, decreasing the slope steepness and allowing TRW formation leads to denser, less diluted DSW; including tides in the moderate and gentle slope cases has no significant influence on the density and tracer distribution. These results indicate that tidal advection reduces entrainment into the DSW layer only when TRWs are suppressed, and that gentler slopes (such as those in the Weddell

Sea) permit DSW to descend to the deep ocean with much less modification than do steeper slopes (such as those in the Ross Sea).

Our findings regarding the importance of tides for overflows over steep slopes raise a further question: how strong must the tides be to substantially impact the descent of DSW? To investigate this, we conduct a series of sensitivity experiments with constant slope steepness ($s = 0.15$) but varying amplitude of the tidal forcing (see "Methods"). Figure 4d shows that the density of the newly formed AABW tends to increase with tidal forcing strength (see Supplementary Fig. 6 for $H_{DSW}$), but more DSW tracer is diluted into lighter waters when tidal flows are sufficiently strong (2K1). A plausible explanation is that tides actually have two competing effects on DSW entrainment: one is to advect DSW quickly to the deep ocean during the ebb tide and reduce the cumulative entrainment, and the other is to produce shear-driven vertical turbulent mixing, which could enhance entrainment when tides are strong. As tidal forcing continues to increase, e.g., to strengths of 3K1 and 4K1, the DSW becomes more diluted than in the experiments without tides and with weaker tides, suggesting that the effect of increased tidal mixing overwhelms that of downslope tidal advection (Supplementary Fig. 7). However, these cases correspond to tidal amplitudes much larger than those observed in the northwestern Ross Sea, where some of the strongest tidal flows were observed around Antarctica (Fig. 5b). Furthermore, both the observations (Fig. 1) and realistic models[32] support that the spring tides favor the formation

of AABW in the northwestern Ross Sea. We therefore infer that tidal flows favor the formation of denser AABW around Antarctica, but future work should further investigate the competing effects of tidal advection and tidal mixing on DSW overflows.

One might also expect that the density difference between DSW and surrounding waters, which controls the baroclinic pressure gradient, influences the process of DSW descent. To address this issue, we conducted sensitivity experiments with different DSW densities while holding the properties of surrounding waters unchanged. To cover the typical dynamical regimes, the experiments include cases with steep ($s = 0.15$), moderate ($s = 0.067$) and small ($s = 0.033$) topographic slopes. The results exhibit no significant differences for the moderate and small slope cases as DSW density changes (Supplementary Fig. 8). For steep slope cases, the tidal induced AABW density increase is not

significant when the DSW is lighter (Supplementary Fig. 8d), which may be explained by weaker interfacial stratification that favors the shear driven mixing under tidal forcing. Overall, varying DSW density does not substantially change the overflow dynamics, suggesting that the results are generally robust over a wide range of DSW density.

Based on above analysis, we can approximately divide the descent of DSW into four different dynamical regimes. When the slope is steep ($s > 0.1$) and tidal forcing is absent, dense overflows take the form of a steady, geostrophic, quasi-along-slope flow and undergo strong mixing; we refer to this as the "Mixing" regime. If there is a substantial tidal flow (here we take the maximum tidal flow speed to be larger than 20 cm/s, see Supplementary Fig. 9), the descent of DSW is accelerated and the entrainment is suppressed; we refer to this as the "Tidal" regime. Over moderate slopes ($0.05 \leq s \leq 0.1$), TRWs are generated,

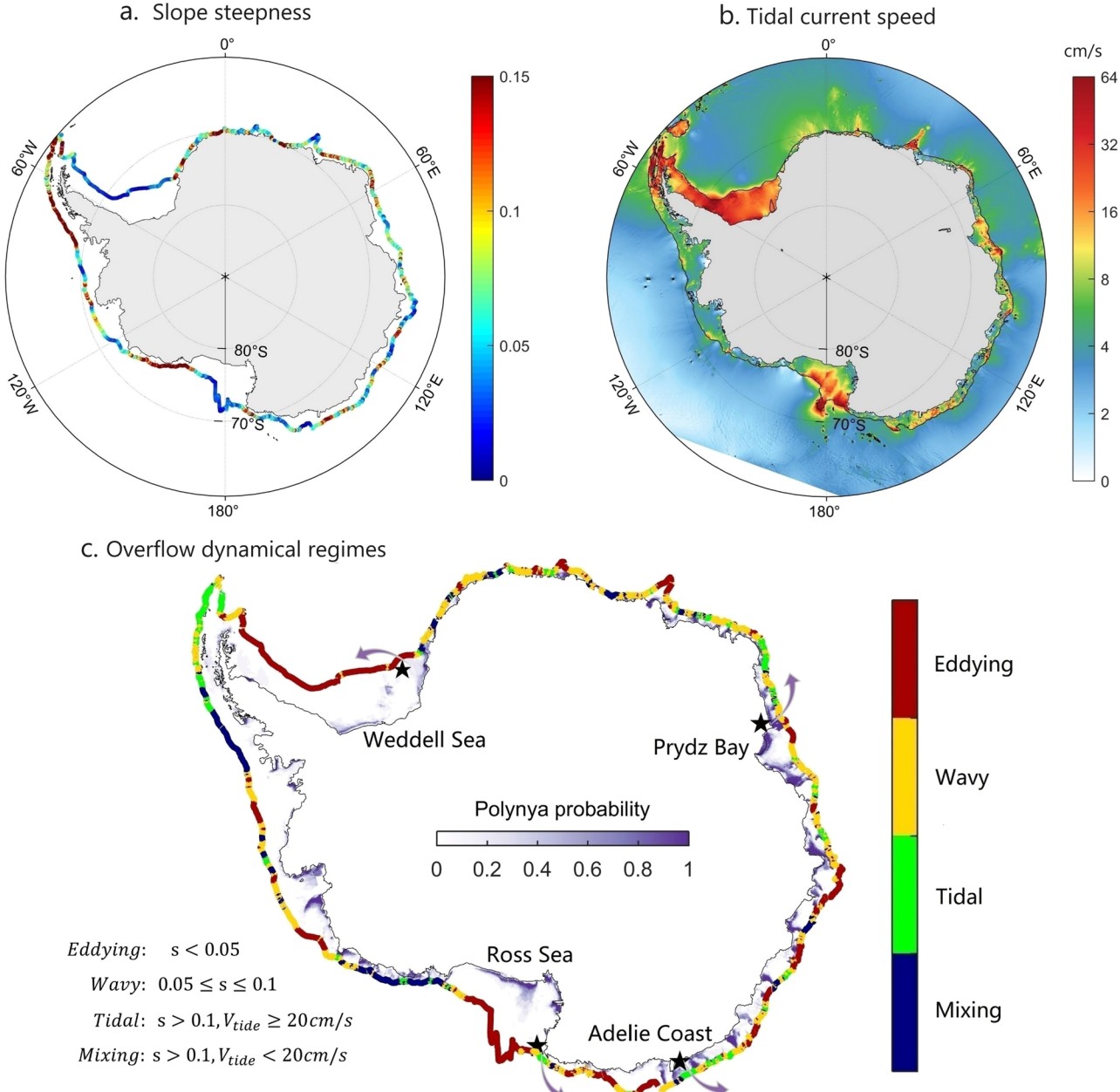

**Fig. 5 | Predicted circum-Antarctic dynamical regimes of dense overflows.**
**a** Averaged continental slope steepness between 800 and 1800 m depth. The topography dataset we use is RTopo-2[62], with 30 arcsec grid spacing. **b** Summation of major axis of four major tidal constituents (O1, K1, M2, S2). **c** Overflow dynamical regimes predicted based on the tidal flow and continental slope steepness. Purple color shading indicates the probability of coastal polynyas occurring between 2013 and 2021, with 1 meaning the polynya occurs every year. The arrows roughly represent the overflow pathway across the continental slope.

accelerating the descent of DSW and suppressing mixing; we refer to this as the "Wavy" regime. When the slope is gentle ($s<0.05$), isolated eddies form on the upper continental slope and suppress the descent of DSW; we refer to this as the "Eddying" regime. Of these four regimes, the "Mixing" regime is least favorable for AABW formation because the combination of a steep slope and weak tides leads to relatively strong entrainment of the overflow (Fig. 4). In contrast, the other three regimes are more favorable for AABW formation. We anticipate relatively fast, localized production of AABW in "Tidal" and "Wavy" regimes, and AABW formation further downstream in the "Eddying" regime due to the slower descent rate associated with the eddy trains over the upper continental slope[16].

In Fig. 5 we apply our classification to estimate overflow dynamical regimes that could arise around the entire Antarctic margins based on the observed slope steepness (Fig. 5a) and tidal flow speeds (Fig. 5b). As seen in Fig. 5c, most of the marginal regions are not in the Mixing regime and thus favorable for AABW formation, provided that DSW are produced in those regions. According to the estimated dynamical regimes in Fig. 5c, DSW overflows in the Adélie coastal region and Prydz Bay should be dominated by tides and TRWs, respectively. Analyses of available moored measurements confirm the estimate in Fig. 5c for these two regions. For the Adélie coastal region (Supplementary Fig. 10), the mooring located on the continental slope records a strong mean flow, superposed with a diurnal tidal cycle of a comparable magnitude, and a fortnightly spring-neap tidal cycle. The temperature records show similar fluctuations, with ebb tides corresponding to colder waters, suggesting that the downslope transport of DSW is mediated by tides. Spectral analysis shows two peaks corresponding to the O1 and K1 tidal constituents[45], but no subtidal TRWs signal. In contrast, the overflows in Prydz Bay are dominated by a subtidal wavy signal, as reported in previous observational studies[13,21]. These are in general consistent with our estimation, thereby confirming the estimate in Fig. 5c.

Since the dense overflows are difficult to detect at the circum-Antarctic scale, and since coastal polynyas can be directly observed via remote sensing and are necessary to produce DSW[46], we plot in Fig. 5c the observed probability of polynya occurrence based on satellite observations during 2013–2021. It is shown that in addition to the four well-known overflow sites, the polynyas are widely spread around Antarctic continental shelf, and the favorable descending regimes in most of these regions suggest that there might be additional sources of AABW, especially in East Antarctica[14]. Note, however, that the presence of polynyas does not guarantee the existence of local DSW varieties, because the continental shelf stratification and circulation may not support the formation of dense water[12,45].

## Discussion

Despite supplying the lower limb of the global meridional overturning circulation, the AABW formation is less well understood than the deep water formation in the northern North Atlantic, partly due to the inhospitable environments around the Antarctic margins[14,47–49]. This study clarifies the roles of tides and TRWs on the formation of AABW, which could serve as a supplement to existing theories of the meridional overturning circulation[50,51]. Based on the results of our high-resolution, process-oriented simulations of dense overflows, we have identified four different dynamical regimes that may occur in overflows around the Antarctic margins. These regimes inform prospective overflow dynamics in different regions, and can be used to infer the places that favor the formation of AABW and guide the overflow observations in the future. The dynamics of tides and TRWs identified in this study are not yet routinely resolved or parameterized in the present climate models, nor are they incorporated into overflow parameterizations[4,52]. Our findings therefore provide a basis for further improvement of ocean and climate models.

One caveat is that the estimated dynamical regimes are sensitive to local slope steepness, and alternate frequently along the continental slope in some places (Fig. 5c). Considering the necessary spatial scale for the growth of waves, variable bathymetry along the continental slope may limit the development of "Wavy" and "Eddying" regimes. Therefore, Fig. 5c could be improved by further observations that, for example, can offer information about the minimum spatial scales allowing the dynamical regimes to fully develop.

A key outcome of this study is that both tides and TRWs can accelerate the descent of DSW and reduce the entrainment of overlying waters. However, tides only substantially influence DSW overflows when the continental slope is steep enough for TRWs to be suppressed. Both tides and TRWs can induce a V-shaped hydrographic front that brings DSW and surface water masses into contact with each other, resulting in AABW that incorporates surface waters into its properties. This implies that changes in surface waters are potentially important to the production of AABW, especially in the context of rapid climate change. However, the present study does not explore the impact of such changes in the interest of simplicity. Our simulations also neglect the influence of the Antarctic Slope Front/Current (ASF/ASC) system, which is associated with cold and fresher water on the continental shelf and a shoreward-deepening pycnocline[41]. Interactions between the ASF/ASC and dense overflows need to be carefully investigated in the future, potentially with the aid of more realistic model simulations. Another implication of our work is that biogeochemical materials such as carbon dioxide and dissolved oxygen could also be transported to the deep ocean more efficiently in Antarctic margins that favor the generation of tides or TRWs, which may contribute to their abyssal storage and thus millennial-scale climate change[7,53].

## Methods
### Observational data
We collected historical in situ observations in the Weddell Sea and Ross Sea, including moorings and cruise data. In the Weddell Sea, we collected data from all 19 available moorings (Fig. 3a) over the period 1968–2011 in vicinity of the overflow. The mooring records are each ~1–2 years in duration. In the Ross Sea, we collected data from the moorings from the 2003-2005 Antarctic Slope (AnSlope) program and the 2007–2011 Cape Adare Long Term Mooring (CALM) program, both of which targeted the overflow in northwestern Ross Sea. Since the moorings are deployed by different observational programs, the recorded hydrographic information and frequency of the recordings are also slightly different. To make full use of these moorings, we omitted the interannual variability and use the jointly recorded hourly temperature and current speed/directions. In addition, as tidal flow can cause significant blow-down of the moorings[23,54], especially for the instruments well above the sea floor, we only used the data from the instruments closest to the sea floor, which is ~20 mab in the Ross Sea and 25–125 mab in the Weddell Sea. In the Ross Sea, we also used the CTD sections from the AnSlope program, to analyze the vertical/cross-slope distribution of temperature and salinity.

To validate the estimation of overflow dynamical regimes around Antarctica, we further collected the mooring observations in the Adélie coastal region. The only mooring on the continental slope is from Mertz Polynya Experiment[45], which records ~500 days flow speed/direction by moored ADCP, and also records temperature by several SBE instruments. We use the velocity records at 1162 m depth (~20 mab) and the temperature records ~10 m above that depth to identify the overflow dynamics.

### Model setup
The numerical model we used is Regional Ocean Modelling System (ROMS), selected for its fidelity in representing oceanic flows over steep slopes[55,56]. The model domain (~1200 km × 650 km) consists of an

embayment (~300 km × 300 km), connected to a flat abyssal ocean of 2500 m depth via a linear slope whose steepness is varied between our experiments. There is also a trough on the western side of the embayment, with a depth of 600 m at the shelf break, increasing to 800 m depth at the south boundary. The bathymetric connection points between the continental slope and continental shelf and abyssal ocean were smoothed by tangent functions. For computational efficiency we used a stretched horizontal grid, with a spacing of 0.5 km near the trough mouth, increasing to ~4 km close to the open boundaries. The vertical discretization uses 60 topography-following levels with increased resolution close to the sea floor (~5 m over the upper slope). The initial stratification is adapted from in situ observations in the Ross Sea[20] (Station 47), with relatively strong stratification in the upper ~200 m and a weaker stratification below (Supplementary Fig. 2). A constant Coriolis parameter of $f = -1.38 \times 10^{-4}$ (72ºS) is used throughout the model domain. Vertical viscosity and mixing are parameterized via the Mellor–Yamada level 2.5 turbulence closure scheme[57]. The benthic stress is parameterized as a quadratic drag with constant drag coefficient of Cd = 0.003.

To simulate a Ross Sea-like overflow, we set the trough width to 30 km and the slope steepness to $s = \frac{\Delta H}{\Delta y} = 0.15$ to approximate the real continental slope along the pathway of the Ross Sea overflows[23]. DSW is restored at the south boundary of the trough (see inset a2 of Supplementary Fig. 2), resulting in an ~0.6 Sv northward flux through the trough that is comparable to the estimated flux in the Ross Sea overflow[20] (~0.8 Sv). The DSW flux is defined by the volume across the continental shelf break, computed over waters with potential densities larger than 1027.86 kg/m³. For the simulation of the Weddell Sea overflow, the trough width is set to 50 km and the slope steepness varies from ~0.07 over the upper continental slope to ~0.02 over the lower continental slope. In contrast to the Ross Sea simulation, DSW is restored to −2.2 °C and 34.73 psu to approximate ice shelf water[2,58]. Here the DSW is denser than the observations in the Weddell Sea[59] and the corresponding flux is ~0.7 Sv, which is somewhat smaller than observed[2] (1.6 ± 0.5 Sv). For the sensitivity experiments presented in Fig. 4, we use similar model setup with the Ross Sea simulation, but changing the slope inclines and using smaller DSW flux (~0.3 Sv) to simulate a range of potential overflow sites around Antarctica.

Tides are forced at the model ocean boundaries via sea surface height fluctuations imposed by extracting the K1 component of the tides in the vicinity of the Ross Sea (Supplementary Fig. 2) from the tidal product TPXO7[60]. Note that the model domain has been scaled relative to its counterpart of the Ross Sea, and the resulting tidal flow in our model is not sensitive to the spatial range of the extracted tidal forcing around the Ross Sea (red sector in the inset a1 of Supplementary Fig. 2).

We first run each model experiment from rest for ~60–100 days without tidal forcing, then branch off two experiments that run for an additional 30 days, with and without tidal forcing. Supplementary Fig. 4 presents the daily averaged overflow fluxes over the last 20 days of 6 different experiments, all of which indicate that the model has reached a quasi-steady state, i.e., the overflow volume fluxes are not exhibiting significant trends over this period. We then analyze the model output data of the last 10 days. Note that it is not realistic to expect complete equilibrium in this process-oriented model setup, which excludes atmospheric forcing. Note also that the differences between the overflow fluxes in our experiments with and without tides are quite variable (Supplementary Table 1); this is due to the changes of circulation and mixing on the continental shelf caused by tidal forcing.

## Isobath-weighted tracer center of mass
The DSW is injected with passive tracers, which can be used to identify the pathway of dense water as it descends to the deep ocean. We track the descent pathway of DSW by computing the tracer-weighted isobath, which quantifies the depth/isobath of the DSW center of mass as a function of along-slope distance:

$$H_{\text{DSW}}(x) = \frac{\overline{\int_0^\infty \int_{-H}^0 H \cdot \tau \, dz dy}}{\int_0^\infty \int_{-H}^0 \tau \, dz dy}. \tag{2}$$

Here $\tau$ and $H$ denote the tracer concentration and the isobath depth, respectively, and the overbar indicates a 10-day average.

## Tidal flow calculation
The tidal flow strength shown in Fig. 1c by the black line is calculated as

$$S_t = \langle |v - \langle v \rangle| \rangle, \tag{3}$$

where the angle brackets indicate daily moving average.

The tidal excursion for diurnal tide can be expressed as

$$L = \int_0^{T/2} v_m \sin\left(\frac{2\pi}{T} t\right) dt, \tag{4}$$

where $T$ indicates the period of diurnal tide, $v_m$ indicates the maximum cross-slope tidal flow speed. We take $v_m$ to be 20 cm/s at 1000 m isobath in the Weddell Sea overflow, and the corresponding tidal excursion is about 5 km.

The maximum tidal flow speed shown in Supplementary Fig. 7 is calculated by first computing

$$V_{\text{tide}} = \sqrt{(\bar{u} - \langle \bar{u} \rangle)^2 + (\bar{v} - \langle \bar{v} \rangle)^2}, \tag{5}$$

where the overbars indicate a vertical average over the upper 200 m of the water column, and the angle brackets indicate daily moving average. Then we calculate the mean value of daily maximum flow speed over 10 days as the maximum tidal flow speed.

## Data availability
The mooring data generated in this study have been deposited in the Figshare database[61]. The model data presented in this article are available on request from X.H. (hxx1119@foxmail.com).

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

## Acknowledgements

This work is supported by grants from the National Natural Science Foundation of China (42227901, 42306252, and 41941007), and by the Independent Research Projects of Southern Marine Science and Engineering Guangdong Laboratory (Zhuhai) (SML2023SP201). A.L.S. was supported by the National Science Foundation under Grants OCE-1751386 and OPP-2023244. X.H. thanks Qinghua Yang and Yichen Lin for their help in collecting polynya data. The authors gratefully acknowledge all the efforts in collecting in situ observational data around Antarctica.

## Author contributions

D.C., X.H. and A.L.S. conceived the study. X.H. analyzed the field data and performed numerical experiments. X.H., A.L.S. and D.C. wrote the paper. X.H., X.L. and A.L.S. contributed to the setup of numerical experiments. X.H., D.C., A.L.S., M.J., X.L., Z.W. and A.L.G. contributed to data interpretation and writing.

## Competing interests

The authors declare no competing interests.
