## [Peer Review File · Nature Communications]

Circum-Antarctic bottom water formation mediated by tides and topographic wavesEditorial Note: Parts of this Peer Review File have been redacted as indicated to remove third-party material where no permission to publish could be obtained.

REVIEWER COMMENTS

Reviewer #1 (Remarks to the Author):

General comments for Authors

In this study, authors discussed the general mechanisms controlling the Dense Shelf Water (DSW) overflow off the continental shelves across continental slopes in the Antarctica. By comparing idealized model simulations and mooring observations in southern Weddell Sea continental slope and over the western Ross Sea continental slope, author characterized two distinct dynamic regimes associated with the topographic organisations and slope steepness, namely, the Topographic Rossby Wave (TRW) regime and tidal regime. Further analysis of the model sensitivity experiments leads to a generalization of the dynamic regimes of the DSW overflow across the Antarctic continental slopes.

I found this study very interesting. The model results show good agreement with the observation in Weddell Sea and Ross Sea – model well captures the preferred frequency bands of DSW overflows as observed. A counterintuitive conclusion – tidal currents, with proper topographic feature and strength, act to suppress the entrainment of ambient water into DSW hence promoting the DSW overflow currents, brings novelty into this work.

I am convinced by the direct comparison between observations and models in Weddell and Ross cases, but I found the generalization requires more robust examinations with the addition of observational evidence. DSW overflow has been observed in various locations other than Weddell and Ross. Authors need to double check the observed DSW overflow in other DSW production areas – e.g., Prydz Bay, Adelie Coast, southwest Weddell Sea continental slope (studies suggested that DSW overflow here is comparable with those from Filchner Depression), where there are sufficient mooring measurements to testify the proposed mechanisms (see figure below). And authors have provided little observational evidence here for locations other than overflow from Filchner Sill and Terra Nova Bay.

[FIGURE REDACTED]

Fig 1 in Amblas and Dowdeswell (2018) - The original figure from which Figure 1a in the manuscript is adapted.

I think that the importance of this study is not just about confirming the dynamic regimes of two mooring sites, which is nicely presented, but more so about this ‘generalization’ of the dynamic regimes over rest of the Antarctic continental slopes. Therefore, I suggested authors to provide more observational evidence from various DSW overflow sites into this work.

Specific comments:

1. Line 57-60: Authors should support the model results from the exploration of parameter space with some observational evidence to ensure the robustness of the pan-Antarctic dynamic regime plot.
2. Line 83-85: Shouldn't the stronger DSW overflow correspond to a colder pulse of temperature as expected? Authors mentioned that the tidal mixing, but cited work Muench et al. (2009) described the mixing and entrainment at the upper bound of DSW associated with tidal current, whereas the temperature at 20mab shown in manuscript marks the lower/core of the DSW overflow. I found this sentence a bit confusing.
3. Line 87-89: Figure S1 subpanel shows the current and temperature record at M1 (further inshore), the temperature minimum corresponds to a neap tide which is somewhat different from Figure 1 WC record. How can this be consistent as claimed by authors?
4. Line 225-229: Authors speculated two competing mechanisms brought by tidal current on DSW descend and entrainment associated with the magnitude of tidal current. Have author considered this into the 'generalization'? Wherever the tidal regime is identified, can the tidal current be strong enough to allow shear-driven mixing to dominate over the tidal advection?
5. Figure 5c: This figure shows that the dynamic regimes are sensitive to the slope steepness. For example, in the Weddell Sea, the overflow sites from the observation sits at the narrow band of 'Wavy' whereas other part of the shelf break is dominated by the 'Eddying' regime. This raises a concern – whether it is needed to categorize the entire shelf break, or should one simply focus on the overflow sites where DSW mostly descends?

Additionally, in Weddell and Ross, if it were without the observation, one can be illuded by this figure saying Weddell and Ross are both 'Eddying'. Authors didn't provide observational evidence for other DSW overflow regions like Adelie Coast, Prydz Bay and southwestern Weddell continental shelf break, and if one tries to identify the exact dynamic regimes using provided figure 5c, it may end up with finding that DSW overflow regime in Adelie Coast is a mixture of 'Tidal' and 'Wavy' with intermittent 'Mixing', and Prydz Bay is 'Wavy' and 'Eddying'. But are these accurate? Therefore, additional evidence from mooring observation is needed here. The SOOS map suggested that there are a few mooring measurements in both Prydz Bay and Adelie Coast that may be able to facilitate the required analysis (see figure below).

SOOS map of SOOS Mooring networks (<http://www.soosmap.aq/>). [FIGURE REDACTED]

6. Line 282-284: Both tides and TRWs accelerate DSW descent and generate V-shaped isopycnal. However, tides and TRWs exhibit distinctively different spectral density in frequency domain. Does the different behaviour on frequency domain of two dynamic phenomena affect the DSW export differently in terms of its cumulative volume transport, which one is more efficient?

Reviewer #2 (Remarks to the Author):

Albeit the mechanisms of tides and topographic Rossby Waves (TRWs) in transforming dense shelf waters (DSW) to Antarctic Bottom Water (AABW) have been identified in previous studies, this work by Han et al. examines systematically the relative roles of tides and TRWs with respect to the steepness of bottom slope using idealized model experiments that represent the Ross Sea and the Weddell Sea, respectively, which leads to the categorization of four regimes (mixing, tidal, wavy, and eddying) around the Antarctic margin. Such a designation could be valuable if confirmed. No doubt that the clarification about the role of tide in promoting rapid descent of DSW over the steep slope and the complementary effect of TRWs over the moderate slope advances our understanding of AABW formation, but the highlight of this study, in my opinion, is Figure 5c. As such I have the following two concerns:

- 1) According to Figure 5c, the large stretches outside of the Ross Sea and the Weddell Sea belong to the eddying regime, while the wavy and tidal regimes exist only near the eastern and western ends of both seas. Does this pattern, or the distributions seen in Figure 5c in general, concur with the “observed/known” locations of AABW formation? In other words, Figure 5c needs to be validated.
- 2) The analyses of DSW are based on tracer concentration rather than its T/S properties. How the temperature and salinity consequently the differential density between DSW and surrounding waters affect the DSW spread (hence the conclusions) is not addressed in the manuscript.

Responses to Reviewers

General response:

We sincerely thank the two reviewers for their constructive comments and suggestions. Their common concern is the validation of Figure 5c by using additional observational evidences at other overflow sites. We now show that the observed overflow dynamical regimes in the Adelie coastal region and Prydz Bay are also consistent with our model-based estimations. The following are our specific responses, with the original comments quoted in *Italic font*, the responses in non-Italic font, and the revised sentences in the manuscript in red font.

Reviewer #1 (Remarks to the Author):

General comments for Authors

In this study, authors discussed the general mechanisms controlling the Dense Shelf Water (DSW) overflow off the continental shelves across continental slopes in the Antarctica. By comparing idealized model simulations and mooring observations in southern Weddell Sea continental slope and over the western Ross Sea continental slope, author characterized two distinct dynamic regimes associated with the topographic organisations and slope steepness, namely, the Topographic Rossby Wave (TRW) regime and tidal regime. Further analysis of the model sensitivity experiments leads to a generalization of the dynamic regimes of the DSW overflow across the Antarctic continental slopes.

I found this study very interesting. The model results show good agreement with the observation in Weddell Sea and Ross Sea – model well captures the preferred frequency bands of DSW overflows as observed. A counterintuitive conclusion – tidal currents, with proper topographic feature and strength, act to suppress the entrainment of ambient water into DSW hence promoting the DSW overflow currents, brings novelty into this work.

I am convinced by the direct comparison between observations and models in Weddell and Ross

cases, but I found the generalization requires more robust examinations with the addition of observational evidence. DSW overflow has been observed in various locations other than Weddell and Ross. Authors need to double check the observed DSW overflow in other DSW production areas – e.g., Prydz Bay, Adelie Coast, southwest Weddell Sea continental slope (studies suggested that DSW overflow here is comparable with those from Filchner Depression), where there are sufficient mooring measurements to testify the proposed mechanisms (see figure below). And authors have provided little observational evidence here for locations other than overflow from Filchner Sill and Tera Nova Bay.

[REDACTED]

Fig 1 in Amblas and Dowdeswell (2018) - The original figure from which Figure 1a in the manuscript is adapted.

I think that the importance of this study is not just about confirming the dynamic regimes of two mooring sites, which is nicely presented, but more so about this ‘generalization’ of the dynamic regimes over rest of the Antarctic continental slopes. Therefore, I suggested authors to provide more observational evidence from various DSW overflow sites into this work.

Response:

Thanks for your very insightful suggestions. We agree that additional observational evidences can make the results in this study more robust. To verify the pan-Antarctic overflow dynamical regimes shown in Fig. 5c, we need the mooring observations on the continental slope,

because the dynamical regimes are estimated based on the slope steepness between 800-1800 m isobaths. This further limit the available moorings in the slope regions around Antarctica. Fortunately, there are a few moorings in the Adelie coastal region and Prydz Bay for validating the overflow dynamical regimes, but no available mooring observations in the slope region of the southwestern Weddell Sea. The following are the evidences observed by the moorings in the Adelie coastal region and Prydz Bay, which further support our conclusions.

Fig. R1a shows the bathymetry of the Adelie coastal region and the site of the only mooring on the continental slope. The corresponding estimated dynamical regimes are shown in Fig. R1b (Zoom in of Fig. 5c), which indicates that the overflow across the mooring site should be dominated by 'Tidal' regime. The ~500 days flow speed/direction at 1162 m depth recorded by lowered ADCP, and the temperature records ~10 m above that depth can be used to validate the overflow dynamical regimes. As shown in Fig. R1c, the northward (approximately cross-slope direction) velocity shows strong mean flow, superimposed with a significant diurnal tidal cycle and a fortnight spring-neap tidal cycle. The temperature shows similar fluctuations, with ebb tides corresponding to colder waters, suggesting that the downslope transport of DSW is mediated by tides. Spectral analysis shows two peaks of O1 and K1 tidal constituents, but no subtidal topographic Rossby waves (TRWs) signal. These are typical features of 'Tidal' regime, which is consistent with our estimation. The similar spectral analysis was presented in Williams et al. (2010).

Figure R1 (Figure S10). Adelie coast overflow observations. (a) Topography of the Adelie coastal region. The red dot indicates the mooring site, and the arrow denotes the export of DSW. (b) The estimated dynamical regimes around the Adelie coastal region. (c) Time series of northward velocity and temperature of selected 45 days. (d) Power spectral density (PSD) of northward velocity and temperature.

For the overflow in Prydz Bay, the mooring observations were reported in Ohshima et al., (2013); however, these data are still not publicly available. Fortunately, the data and the analysis of these data had been shown in several published papers, which provide the necessary information that could validate the overflow dynamical regime in the Prydz Bay overflow region.

The mooring sites are shown in Fig. R2a. The dynamical regimes along the continental slope of Cape Darnley are shown in Fig. R2b, which are primarily dominated by ‘Wavy’ and ‘Eddying’, but with intermittent ‘Mixing’ and ‘Tidal’ regimes at the western side. Note the estimated dynamical regimes are based on local slope steepness between 800 m and 1800 m isobaths, while moorings M1, M3 and M4 are in deeper ocean with less steep slope. Therefore, the dynamical regimes for moorings M1-M4 are prone to be ‘Wavy’ or ‘Eddying’. Fig. R2c shows the time series of flow speed/direction at moorings M1, M2 and M3, all exhibiting

fluctuations of several days. Power spectral density (PSD) analysis clarify the periods of fluctuations (Fig. R2d), and further identify the fluctuations to be TRWs by numerical models (Nakayama et al., 2014). Note that we cannot distinguish the ‘Wavy’ and ‘Eddying’ regimes based on the sparse overflow current observations, because they all take the form of oscillations. Nevertheless, these observational results are in general consistent with our classifications of overflow dynamical regimes in this region.

[REDACTED]

Figure R2. Mooring observations in the Cape Darnley (Prydz Bay) overflow region. (a) The bathymetry is indicated by blue contours. Mooring locations are indicated by orange symbols, with the mean velocity at the bottom and top of the mooring shown as orange and yellow vectors, respectively. This figure is Figure 1 of Ohshima et al., (2013). (b) The estimated dynamical regimes along the continental slope of Cape Darnley. (c) M1 velocity vector at 92 m above the bottom. For M2 and M3 moorings, the curves indicate velocity component of the mean (dominant) flow direction at 20 m (blue) and 226 m (red) above the bottom. This figure is Figure S3 of Ohshima et al., (2013). (d) Fourier spectra of moorings M1 and M3. This figure is Figure 5a of Nakayama et al., (2014).

We include Figure R1 in the revised manuscript as Figure S10, and describe these further validations in the main text in Lines 278–289 in the revised manuscript: “According to the estimated dynamical regimes in Fig. 5c, DSW overflows in the Adelie coastal region and Prydz Bay should be dominated by tides and TRWs, respectively. Analysis of available moored

measurements confirm the estimate in Fig. 5c for these two regions. For the Adelie coastal region (Fig. S10), the mooring that is located on the continental slope records a strong mean flow, upon which is superposed a diurnal tidal cycle of a comparable magnitude, with a fortnightly spring-neap tidal cycle. The temperature records show similar fluctuations, with ebb tides corresponding to colder waters, suggesting that the downslope transport of DSW is mediated by tides. Spectral analysis shows two peaks corresponding to the O1 and K1 tidal constituents⁴⁶, but no subtidal TRWs signal. In contrast, the overflows in Prydz Bay are dominated by a subtidal wavy signal, as reported in previous observational studies^{13,21}. These are in general consistent with our estimation, thereby confirming the estimate in Fig. 5c.”

Specific comments:

1. Line 57-60: Authors should support the model results from the exploration of parameter space with some observational evidence to ensure the robustness of the pan-Antarctic dynamic regime plot.

Response:

As shown above, we have collected all the available mooring data that can be used to verify our conclusion. Although the available moorings are sparse, these additional evidences have increased the robustness of the pan-Antarctic dynamic regime plot.

2. Line 83-85: Shouldn't the stronger DSW overflow correspond to a colder pulse of temperature as expected? Authors mentioned that the tidal mixing, but cited work Muench et al. (2009) described the mixing and entrainment at the upper bound of DSW associated with tidal current, whereas the temperature at 20mab shown in manuscript marks the lower/core of the DSW overflow. I found this sentence a bit confusing.

Response:

Thanks for this very insightful comment. Yes, mooring observations on the upper continental slope show that strong tidal flows correspond to colder DSW pulses (Whitworth and Orsi, 2006), and numerical experiments further suggested that the colder pulses can reach

deep ocean (Wang et al., 2010). In this study, we synthesize all available historical mooring data in the northwestern Ross Sea to verify that tidal advection is dominant on controlling DSW descent, and produce colder and denser AABW.

The dilution of dense overflow can be caused by interfacial (upper bound of dense overflow) shear-driven mixing and bottom stress induced mixing, which are both important to modifying the outflow hydrographic properties. As discussed in Peters et al., (2005), the interfacial shear-driven mixing cannot directly reflect the property changes of bottom layer waters, because of the large thickness of overflow plumes. Besides the interfacial mixing, Muench et al., (2009) also emphasized that the strong benthic stress induced by a spring tide can augment strong mixing to the upper interface of dense overflows, which is able to dilute the outflow efficiently. We think that these results support our argument to some extent, so we cited this paper.

To avoid potential confusion, we have changed reference 32 (Muench et al., 2009) to Wang et al., (2010) and delete this sentence and rewrite the next sentence (Lines 83-86 in the revised manuscript): “**The fact that the coldest waters emerge during the spring tide therefore suggests that temperature variability is governed by the tendency of the tides to advect DSW across the continental slope^{22,23}, rather than by the mixing induced by the tidal flow^{32,33}.**”

3. Line 87-89: Figure S1 subpanel shows the current and temperature record at M1 (further inshore), the temperature minimum corresponds to a neap tide which is somewhat different from Figure 1 WC record. How can this be consistent as claimed by authors?

Response:

Thanks for pointing out this problem. We agree that we did not clarify this clearly in the manuscript. Actually, tidal effects on dense overflows on the continental shelf are different from those on the continental slope, i.e., tidal flows dilute DSW on the continental shelf (moorings M1 and CA), while decreasing net entrainment on the continental slope. The realistic numerical experiments showed similar results (Figure 9 in Wang et al., 2010).

Here we provide a possible explanation: Without tidal forcing, overflow speed is weak on the flat continental shelf but strong on the continental slope due to gravity acceleration, and the corresponding shear-driven mixing is thus weak and strong respectively; When adding tides, it

theoretically increases mixing throughout the water column, which is the situation on the continental shelf; In contrast, tidal advection accelerates DSW downslope descent, decreasing net entrainment on the continental slope.

We add an additional description in the main text (Lines 88-89 in the revised manuscript): “In contrast, a stronger tidal flow dilutes DSW on the continental shelf break (moorings M1 and CA, Fig. S1), consistent with the influence of stronger tidal mixing^{32,33}.” And in Lines 94-96 in the revised manuscript: “The results show that in a Ross Sea parameter regime, tracers are advected to greater depths and retain higher concentration in the presence of tides (Figs. 2a,b), though the tracer concentrations are more diluted in the regions shallower than 500 m depth (Figs. 2c,d).”

4. Line 225-229: Authors speculated two competing mechanisms brought by tidal current on DSW descend and entrainment associated with the magnitude of tidal current. Have author considered this into the ‘generalization’? Wherever the tidal regime is identified, can the tidal current be strong enough to allow shear-driven mixing to dominate over the tidal advection?

Response:

Yes, it is very interesting to generalize the impacts of tidal flow strength on dense overflow mixing. Based on our sensitivity experiments shown in the manuscript, the maximum density of bottom water increases as the tidal flow strength increases. However, more tracers are diluted into lighter water in 2K1 experiment, and obviously the mean density decreases relative to 1K1 experiment (Fig. R3). This suggests that the effect of tidal mixing increases relative to tidal advection for 2K1 tidal experiment, and we think that the effect of tidal mixing will continuously increase and dilute dense overflow further under stronger tidal forcing. Therefore, we run another two experiments with extremely strong tidal forcing (3K1 and 4K1) to investigate this further. The results show that the DSW is more diluted than the experiments without tides and with weaker tides, which indicates that tidal mixing completely overwhelms tidal advection in 3K1 and 4K1 experiments.

Figure R3 (Figure S7). Probability density of volume-integrated tracer mass versus fluid potential density for experiments with different tidal forcing but constant slope steepness ($s = 0.15$).

However, the 2K1 tidal forcing already produces a very strong tidal flow (> 0.5 m/s) that is comparable to the observed strongest tidal flow around Antarctica. Therefore, we think that the conclusion in this study is suitable to circum-Antarctica conditions. To fully generalize the impacts of tides on AABW properties, we need to investigate several potential factors comprehensively by using a realistic model, e.g., dense overflow flux, tidal flow strength, the continental slope steepness, and so on. The impacts of these factors need to be investigated in details and are out of scope of this study. Therefore, we think that these will need to be done in the future.

We have included Figure R3 in the revised manuscript as Figure S7, and added more discussion in the main text (Lines 231-239 in the revised manuscript): “As tidal forcing continues to increase, e.g., to strengths of 3K1 and 4K1, the DSW becomes more diluted than in the experiments without tides and with weaker tides, suggesting that the effect of increased tidal mixing overwhelms that of downslope tidal advection (Fig. S7). However, these cases correspond to tidal amplitudes much larger than those observed in the northwestern Ross Sea, where some of the strongest tidal flows were observed around Antarctica (Fig. 5b). Furthermore, both the observations (Fig. 1) and realistic models³² support that the spring tides favor the formation of AABW in the northwestern Ross Sea. We therefore infer that tidal flows favor the formation of denser AABW around Antarctica, but future work should further investigate the competing effects of tidal advection and tidal mixing on DSW overflows.”

5. *Figure 5c: This figure shows that the dynamic regimes are sensitive to the slope steepness. For example, in the Weddell Sea, the overflow sites from the observation sits at the narrow band of 'Wavy' whereas other part of the shelf break is dominated by the 'Eddying' regime. This raises a concern – whether it is needed to categorize the entire shelf break, or should one simply focus on the overflow sites where DSW mostly descends?*

Response:

Thanks for pointing this out. We agree that the overflow dynamical regimes are sensitive to the local slope steepness. To date there have been only several well identified overflow sites, including the Ross Sea, Weddell Sea, Prydz Bay and Adelie coastal region. However, more and more overflow sites are being discovered as observations increase (Amblas and Dowdeswell, 2018), that could potentially contribute to AABW formation. Therefore, we estimate the overflow dynamical regimes of the entire shelf break, which could be used to infer the places that favor the formation of AABW and guide the overflow observations in the future.

Additionally, in Weddell and Ross, if it were without the observation, one can be illuded by this figure saying Weddell and Ross are both 'Eddying'. Authors didn't provide observational evidence for other DSW overflow regions like Adelie Coast, Prydz Bay and southwestern Weddell continental shelf break, and if one tries to identify the exact dynamic regimes using provided figure 5c, it may end up with finding that DSW overflow regime in Adelie Coast is a mixture of 'Tidal' and 'Wavy' with intermittent 'Mixing', and Prydz Bay is 'Wavy' and 'Eddying'. But are these accurate? Therefore, additional evidence from mooring observation is needed here. The SOOS map suggested that there are a few mooring measurements in both Prydz Bay and Adelie Coast that may be able to facilitate the required analysis (see figure below).

[REDACTED]

SOOS map of SOOS Mooring networks (<http://www.soosmap.aq/>).

Response:

Thank you very much for the suggestions and the useful information for mooring networks. The DSW overflows across the continental shelf through narrow troughs with the horizontal scale on the order of 10 kilometers, so the overflow dynamical regimes are sensitive to local slope steepness. In Figure 5c, we estimate the dynamical regimes by local slope steepness without doing any along-slope average or smoothing, and thus the overflow dynamical regimes alternate frequently in some places (e.g., Fig. R2c) due to variable bathymetry. Considering the necessary spatial scale for the growth of waves, this may limit the development of the ‘Wavy’ and ‘Eddying’ regimes. Although the dynamical regimes are variable, the observed overflow dynamical regimes are consistent with our estimation, as shown in Fig. R1 and Fig. R2.

Currently, Fig. 5c is a rough estimation of circum-Antarctica overflow dynamical regimes, which need to be improved as observations increase, especially for identifying the minimum spatial scales allowing the dynamical regimes to fully develop. We have added more discussion in the main text (Lines 312-317 in the revised manuscript): “One caveat is that the estimated dynamical regimes are sensitive to local slope steepness, and alternate frequently along the continental slope in some places (Fig. 5c). Considering the necessary spatial scale for the growth of waves, variable bathymetry along the continental slope may limit the development of ‘Wavy’ and ‘Eddying’ regimes. Therefore, Fig. 5c could be improved by further observations that, for example, can offer information about the minimum spatial scales allowing the dynamical regimes to fully develop.”

6. Line 282-284: *Both tides and TRWs accelerate DSW descent and generate V-shaped isopycnal. However, tides and TRWs exhibit distinctively different spectral density in frequency domain. Does the different behaviour on frequency domain of two dynamic phenomena affect the DSW export differently in terms of its cumulative volume transport, which one is more efficient?*

Response:

Tides and TRWs can both modulate DSW export in the form of periodic downslope flows, but their frequencies are different, which potentially influence the efficiencies of DSW downslope transport.

As shown by the pathway of the tracer center of mass (Fig. R4), tidal modulated DSW transport is only important for steep slopes where there are no TRWs, and becomes almost negligible when TRWs occur for gentle slopes. These are consistent with our general understanding that tides are driven by external forces and advect DSW across the continental slope, and thus the tidal flow speed (excursion) and the width of continental slope are important for DSW downslope transport. For less steep slopes, the continental slope is wider because of the constant depth of abyssal ocean in our model setup, while tidal flow is weaker under a constant tidal forcing. Therefore, tidal excursion is relatively small compared to the width of the continental slope for a gentle slope. In contrast, the mechanism of TRWs on DSW transport is that the baroclinic instability of dense overflows release potential energy to energize TRWs by sliding into deep ocean. Therefore, DSW can continuously slide into deep ocean for gentle slopes where TRWs occur. The mathematical model for DSW descent rate has been presented in Han et al. (2023), which suggests that the transport efficiencies do not rely on the frequencies of TRWs.

Overall, the differences of tides and TRWs on DSW downslope transport are due to their different dynamics, rather than the differences of frequency domain. Tides and TRWs dominate the DSW transport for steep and gentle slopes respectively, which are the basements of our clarification of ‘Tidal’ and ‘Wavy’ regimes.

Figure R4 (revised Figure 4a,b in the manuscript). Simulations investigating the influences of tidal flows and continental slope steepness on DSW descent. (a) Isobath corresponding to the DSW tracer center of mass, as a function of along-slope distance downstream of the trough. The solid thick curves indicate the simulations without tidal forcing, while the dashed curves indicate the corresponding simulations that include tidal forcing. The semi-transparent red shading denotes the zonal range that we calculate the averaged DSW tracer-weighted isobath shown in panel b. (b) Zonally-averaged DSW tracer-weighted isobath for different experiments, with and without tidal forcing.

References:

Amblas, D., & Dowdeswell, J. A. (2018). Physiographic influences on dense shelf-water cascading down the Antarctic continental slope. *Earth-Science Reviews*, 185, 887–900.

<https://doi.org/10.1016/j.earscirev.2018.07.014>

Han, X., Stewart, A. L., Chen, D., Liu, X., & Lian, T. (2023). Controls of topographic Rossby wave properties and downslope transport in dense overflows. *Journal of Physical Oceanography*.

<https://doi.org/10.1175/JPO-D-22-0237.1>

Muench, R., Padman, L., Gordon, A., & Orsi, A. (2009). A dense water outflow from the Ross Sea, Antarctica: Mixing and the contribution of tides. *Journal of Marine Systems*, 77(4), 369-387. <https://doi.org/10.1016/j.jmarsys.2008.11.003>

Nakayama, Y., Ohshima, K. I., Matsumura, Y., Fukamachi, Y., & Hasumi, H. (2014). A numerical investigation of formation and variability of Antarctic Bottom Water off Cape Darnley, East Antarctica. *Journal of Physical Oceanography*, 44(11), 2921-2937. <https://doi.org/10.1175/JPO-D-14-0069.1>

Ohshima, K. I., Fukamachi, Y., Williams, G. D., Nihashi, S., Roquet, F., Kitade, Y., Tamura, T., Hirano, D., Herraiz-Borreguero, L., Field, I., Hindell, M., Aoki, S., & Wakatsuchi, M. (2013). Antarctic Bottom Water production by intense sea-ice formation in the Cape Darnley polynya. *Nature Geoscience*, 6(3), 235–240. <https://doi.org/10.1038/ngeo1738>

Peters, H., Johns, W. E., Bower, A. S., & Fratantoni, D. M. (2005). Mixing and entrainment in the Red Sea outflow plume. Part I: Plume structure. *Journal of Physical Oceanography*, 35(5), 569-583. <https://doi.org/10.1175/JPO2679.1>

Whitworth, I., & Orsi, A. H. (2006). Antarctic Bottom Water production and export by tides in the Ross Sea. *Geophysical Research Letters*, 33(12). <https://doi.org/10.1029/2006GL026357>

Wang, Q., Danilov, S., Hellmer, H. H., & Schröter, J. (2010). Overflow dynamics and bottom water formation in the western Ross Sea: Influence of tides. *Journal of Geophysical Research: Oceans*, 115(10), 1–16. <https://doi.org/10.1029/2010JC006189>

Williams, G. D., Aoki, S., Jacobs, S. S., Rintoul, S. R., Tamura, T., & Bindoff, N. L. (2010). Antarctic bottom water from the Adélie and George V Land coast, East Antarctica (140–149 E). *Journal of Geophysical Research: Oceans*, 115(C4). <https://doi.org/10.1029/2009JC005812>

Reviewer #2 (Remarks to the Author):

Albeit the mechanisms of tides and topographic Rossby Waves (TRWs) in transforming dense shelf waters (DSW) to Antarctic Bottom Water (AABW) have been identified in previous studies, this work by Han et al. examines systematically the relative roles of tides and TRWs with respect to the steepness of bottom slope using idealized model experiments that represent the Ross Sea and the Weddell Sea, respectively, which leads to the categorization of four regimes (mixing, tidal, wavy, and eddying) around the Antarctic margin. Such a designation could be valuable if confirmed. No doubt that the clarification about the role of tide in promoting rapid descent of DSW over the steep slope and the complementary effect of TRWs over the moderate slope advances our understanding of AABW formation, but the highlight of this study, in my opinion, is Figure 5c. As such I have the following two concerns:

1) According to Figure 5c, the large stretches outside of the Ross Sea and the Weddell Sea belong to the eddying regime, while the wavy and tidal regimes exist only near the eastern and western ends of both seas. Does this pattern, or the distributions seen in Figure 5c in general, concur with the “observed/known” locations of AABW formation? In other words, Figure 5c needs to be validated.

Response:

We sincerely thank the reviewer for this very helpful suggestion, which is also the concern of another reviewer. To verify the pan-Antarctic overflow dynamical regimes shown in Fig. 5c, we need the mooring observations on the continental slope, because the dynamical regimes are estimated based on the slope steepness between 800-1800 m isobaths. This further reduces the number of the available moorings around Antarctica. Fortunately, there are a few moorings in the Adelie coastal region and Prydz Bay, which can be used to validate Fig. 5c.

Fig. R1a shows the bathymetry of the Adelie coastal region and the site of the only mooring on the continental slope. The corresponding estimated dynamical regimes are shown in Fig. R1b (Zoom in of Fig. 5c), which indicates that the overflow across the mooring site should be dominated by ‘Tidal’ regime. The ~500 days flow speed/direction at 1162 m depth

recorded by lowered ADCP, and the temperature records ~ 10 m above that depth can be used to validate the overflow dynamical regimes. As shown in Fig. R1c, the northward (approximately cross-slope direction) velocity shows strong mean flow, superimposed with a significant diurnal tidal cycle and a fortnight spring-neap tidal cycle. The temperature shows similar fluctuations, with ebb tides corresponding to colder waters, suggesting that the downslope transport of DSW is mediated by tides. Spectral analysis shows two peaks of O_1 and K_1 tidal constituents, but no subtidal topographic Rossby waves (TRWs) signal. These are typical features of ‘Tidal’ regime, which is consistent with our estimation. The similar spectral analysis was presented in Williams et al. (2010).

Figure R1 (Figure S10). Adelie coast overflow observations. (a) Topography of the Adelie coastal region. The red dot indicates the mooring site, and the arrow denotes the export of DSW. (b) The estimated dynamical regimes around the Adelie coastal region. (c) Time series of northward velocity and temperature of selected 45 days. (d) Power spectral density (PSD) of northward velocity and temperature.

For the overflow in Prydz Bay, the mooring observations were reported in Ohshima et al., (2013); however, these data are still not publicly available. Fortunately, the data and the analysis of these data had been shown in several published papers, which provide the necessary

information that could validate the overflow dynamical regime in the Prydz Bay overflow region.

The mooring sites are shown in Fig. R2a. The dynamical regimes along the continental slope of Cape Darnley are shown in Fig. R2b, which are primarily dominated by ‘Wavy’ and ‘Eddying’, but with intermittent ‘Mixing’ and ‘Tidal’ regimes at the western side. Note the estimated dynamical regimes are based on local slope steepness between 800 m and 1800 m isobaths, while moorings M1, M3 and M4 are in deeper ocean with less steep slope. Therefore, the dynamical regimes for moorings M1-M4 are prone to be ‘Wavy’ or ‘Eddying’. Fig. R2c shows the time series of flow speed/direction at moorings M1, M2 and M3, all exhibiting fluctuations of several days. Power spectral density (PSD) analysis clarify the periods of fluctuations (Fig. R2d), and further identify the fluctuations to be TRWs by numerical models (Nakayama et al., 2014). Note that we cannot distinguish the ‘Wavy’ and ‘Eddying’ regimes based on the sparse overflow current observations, because they all take the form of oscillations. Nevertheless, these observational results are in general consistent with our classifications of overflow dynamical regimes in this region.

[REDACTED]

Figure R2. Mooring observations in the Cape Darnley (Prydz Bay) overflow region. (a) The bathymetry is indicated by blue contours. Mooring locations are indicated by orange symbols, with the mean velocity at the bottom and top of the mooring shown as orange and yellow vectors, respectively. This figure is

Figure 1 of Ohshima et al., (2013). (b) The estimated dynamical regimes along the continental slope of Cape Darnley. (c) M1 velocity vector at 92 m above the bottom. For M2 and M3 moorings, the curves indicate velocity component of the mean (dominant) flow direction at 20 m (blue) and 226 m (red) above the bottom. This figure is Figure S3 of Ohshima et al., (2013). (d) Fourier spectra of moorings M1 and M3. This figure is Figure 5a of Nakayama et al., (2014).

We include Figure R1 in the revised manuscript as Figure S10, and describe these further validations in the main text in Lines 278–289 in the revised manuscript: “**According to the estimated dynamical regimes in Fig. 5c, DSW overflows in the Adelie coastal region and Prydz Bay should be dominated by tides and TRWs, respectively. Analysis of available moored measurements confirm the estimate in Fig. 5c for these two regions. For the Adelie coastal region (Fig. S10), the mooring that is located on the continental slope records a strong mean flow, upon which is superposed a diurnal tidal cycle of a comparable magnitude, with a fortnightly spring-neap tidal cycle. The temperature records show similar fluctuations, with ebb tides corresponding to colder waters, suggesting that the downslope transport of DSW is mediated by tides. Spectral analysis shows two peaks corresponding to the O1 and K1 tidal constituents⁴⁶, but no subtidal TRWs signal. In contrast, the overflows in Prydz Bay are dominated by a subtidal wavy signal, as reported in previous observational studies^{13,21}. These are in general consistent with our estimation, thereby confirming the estimate in Fig. 5c.**”

2) The analyses of DSW are based on tracer concentration rather than its T/S properties. How the temperature and salinity consequently the differential density between DSW and surrounding waters affect the DSW spread (hence the conclusions) is not addressed in the manuscript.

Response:

Thanks for your very insightful suggestions. The density differences between DSW and surrounding waters potentially influence our conclusions. Here, we carried out additional experiments with different DSW density, but holding the properties of surrounding waters unchanged. To cover the typical dynamical regimes, the experiments include steep, moderate and small topographic slopes. We vary the density of DSW by restoring salinity to different

values, with constant potential temperature of -1.9°C . Since the maximum salinity of the DSW presented in the manuscript is 34.8 psu (Fig. S2), we restore the maximum salinity to 34.7 and 35 psu to simulate lighter and denser DSW respectively.

The results show that the downslope transport of DSW exhibits no significant differences for moderate and small slope experiments as DSW density changes (Figs. R5b,c). For the steep slope experiments without tides (Fig. R4a), the descent of DSW is slightly more efficient as the DSW density increases, owing to the farther offshore travels of the denser DSW during the geostrophic adjustment process. The effects of tides on DSW descent show no substantial differences: tidal advection facilitates DSW descent for steep slopes, while no significant influences for gentle slopes when TRWs or eddies form. As expected, the AABW density increases as the DSW density increases, and the tides have no substantial influences on density distributions for moderate and small slope experiments (Figs. R4e,f). For steep slope experiments (Fig. R4d), tidal induced density increase is not significant when the DSW is lighter, which may be explained by weaker interfacial stratification that favors shear instability under tidal forcing.

Overall, varying DSW density does not change the results substantially, suggesting that our conclusion is generally robust and appropriate for a pan-Antarctica description of AABW formation.

We have added some discussion in the main text (Lines 240-250 in the revised manuscript):
“One might also expect that the density difference between DSW and surrounding waters, which controls the baroclinic pressure gradient, influences the process of DSW descent. To address this issue, we conducted sensitivity experiments with different DSW densities while holding the properties of surrounding waters unchanged. To cover the typical dynamical regimes, the experiments include cases with steep ($s = 0.15$), moderate ($s = 0.067$) and small ($s = 0.033$) topographic slopes. The results exhibit no significant differences for the moderate and small slope cases as DSW density changes (Fig. S8). For steep slope cases, the tidal induced AABW density increase is not significant when the DSW is lighter (Fig. S8d), which may be explained by weaker interfacial stratification that favors the shear driven mixing under tidal forcing. Overall, varying DSW density does not substantially change the overflow dynamics, suggesting that the results are generally robust over a wide range of DSW density.”

Figure R5 (Figure S8 in the revised manuscript). The impact of the density difference between DSW and surrounding waters on the overflow properties. We vary the DSW density by restoring the salinities on the continental shelf to different maximum values, with a constant potential temperature of -1.9°C . The red, green and blue curves correspond to the DSW with maximum salinities of 34.7, 34.8 and 35 psu respectively. The solid thick curves indicate the simulations without tidal forcing, while the dashed curves indicate the corresponding simulations that include tidal forcing. (a-c) Isobath corresponding to the DSW tracer center of mass, as a function of along-slope distance downstream of the trough. The top to bottom panels correspond to steep ($s=0.15$), moderate ($s=0.067$) and small ($s=0.033$) slopes, respectively. (d-f) Probability density function of tracer as a function of potential density, computed over the area downstream of the semi-transparent red shading shown in Fig. 4a.

References:

- Nakayama, Y., Ohshima, K. I., Matsumura, Y., Fukamachi, Y., & Hasumi, H. (2014). A numerical investigation of formation and variability of Antarctic Bottom Water off Cape Darnley, East Antarctica. *Journal of Physical Oceanography*, *44*(11), 2921-2937. <https://doi.org/10.1175/JPO-D-14-0069.1>
- Ohshima, K. I., Fukamachi, Y., Williams, G. D., Nihashi, S., Roquet, F., Kitade, Y., Tamura, T., Hirano, D., Herraiz-Borreguero, L., Field, I., Hindell, M., Aoki, S., & Wakatsuchi, M. (2013). Antarctic Bottom Water production by intense sea-ice formation in the Cape Darnley polynya. *Nature Geoscience*, *6*(3),

235–240. <https://doi.org/10.1038/ngeo1738>

Williams, G. D., Aoki, S., Jacobs, S. S., Rintoul, S. R., Tamura, T., & Bindoff, N. L. (2010). Antarctic bottom water from the Adélie and George V Land coast, East Antarctica (140–149 E). *Journal of Geophysical Research: Oceans*, 115(C4). <https://doi.org/10.1029/2009JC005812>

REVIEWERS' COMMENTS

Reviewer #1 (Remarks to the Author):

The revision was completed comprehensively. The authors included additional mooring observation to support their model results with good consistency. They added extra explanations over ambiguous/confusing places here and there as pointed out in the first review report. Statements are appended in the revised text where the circum-Antarctic dynamic regime map is discussed, this is important for readers to view this 'extrapolation' with caution, which is good. Therefore, I am happy to suggest the acceptance of this revised manuscript. Well done!

Reviewer #2 (Remarks to the Author):

The authors have addressed my questions in the previous round of review by adding analyses and discussions to further support Fig. 5c, as well as another set of sensitivity experiments to demonstrate the effects of relative density between the DSW and the background water. It is interesting to see the discussion about the localness of designated regimes (lines 329-334), which brings up an issue that Fig. 4a seems to suggest there are already large differences at an along-shore distance of 620 km (i.e., in the trough). As such it would be interesting to show the DSW evolution/processes in the near field among these experiments and how the near field is related to the downstream development.

Specific points:

1. Lines 19-20: This study did not discuss the resulting AABW properties. What are the implications of the three (6 considering tides) cases in Fig. 4c regarding the AABW properties? Does it mean that the AABW has the highest density in the "eddying" regime, but it resides in the depth range of 1000-1200m, not reaching the lower slope?
2. Line 292: Change "Analysis" to "Analyses".
3. Rewrite the sentence in lines 293-296.

Reviewer #2 (Remarks to the Author):

The authors have addressed my questions in the previous round of review by adding analyses and discussions to further support Fig. 5c, as well as another set of sensitivity experiments to demonstrate the effects of relative density between the DSW and the background water. It is interesting to see the discussion about the localness of designated regimes (lines 329-334), which brings up an issue that Fig. 4a seems to suggest there are already large differences at an along-shore distance of 620 km (i.e., in the trough). As such it would be interesting to show the DSW evolution/processes in the near field among these experiments and how the near field is related to the downstream development.

Specific points:

- 1. Lines 19-20: This study did not discuss the resulting AABW properties. What are the implications of the three (6 considering tides) cases in Fig. 4c regarding the AABW properties? Does it mean that the AABW has the highest density in the “eddying” regime, but it resides in the depth range of 1000-1200m, not reaching the lower slope?*

Response:

Thanks for your very insightful comments. Fig. 4c indeed implies that the AABW has the highest density in the ‘eddying’ regime, because gentle slopes cause slower geostrophic flow (Nof, 1983), thus weaker shear-driven mixing. Considering that we set constant DSW density over a range of sensitivity experiments, we did not use the words like ‘denser’ or ‘lighter’ to describe newly formed AABW to avoid potential confusions, especially for the Ross- and Weddell-like experiments. The DSW densities in these two regions are different, which contributes to making differences in the resulted AABW densities. Instead, we emphasize the effects of the net entrainment during the DSW descending process on the resulted AABW properties. We have some discussion in the manuscript (Lines 175-178): “These results indicate that tidal advection reduces entrainment into the DSW layer only when TRWs are suppressed, and that gentler slopes (such as those in the Weddell Sea) permit DSW to descend to the deep ocean with much less modification than do steeper slopes (such as those in the Ross Sea).”

Although the descent rate is slower in ‘eddying’ regime, the dense water can gradually descend to deeper ocean as flowing further downstream (Fig. 4a). The related statement lies in the manuscript

(Lines 220-223): ” We anticipate relatively fast, localized production of AABW in ‘Tidal’ and ‘Wavy’ regimes, and AABW formation further downstream in the ‘Eddying’ regime due to the slower descent rate associated with the eddy trains over the upper continental slope¹⁶.”

Fig. 4 | Simulations investigating the influences of tidal flows and continental slope steepness on dense overflows. a Isobath corresponding to the dense shelf water (DSW) tracer center of mass, as a function of along-slope distance downstream of the trough. The solid thick curves indicate the simulations without tidal forcing, while the dashed curves indicate the corresponding simulations that include tidal forcing. The semi-transparent red shading denotes the zonal range that we calculate the averaged DSW tracer-weighted isobath shown in panel **b**. **b** Zonally-averaged DSW tracer-weighted isobath for different experiments, with and without tidal forcing. **c** Probability density function of tracer as a function of potential density for experiments with three different slope inclines, with and without tidal forcing, computed over the area downstream of the semi-transparent red shading shown in panel **a**. The orange vertical line indicates the maximum potential density in the simulation prior to DSW production. **d** Similar to panel **c**, but for varying tidal forcing strengths with constant slope steepness ($s = 0.15$).

2. *Line 292: Change “Analysis” to “Analyses”.*

Corrected.

3. *Rewrite the sentence in lines 293-296.*

Response:

Thanks for your careful reading of this manuscript. We agree that there are some problems of that sentence, so we have now rewritten that sentence to “For the Adelie coastal region (Supplementary Fig. 10), the mooring located on the continental slope records a strong mean flow, superposed with a diurnal tidal cycle of a comparable magnitude, and a fortnightly spring-neap tidal cycle.”

Reference:

Nof, D. (1983). The translation of isolated cold eddies on a sloping bottom. *Deep Sea Research Part A. Oceanographic Research Papers*, 30(2), 171-182.